# Phototriggered protein syntheses by using (7-diethylaminocoumarin-4-yl)methoxycarbonyl-caged aminoacyl tRNAs

Takashi Ohtsuki[1], Shigeto Kanzaki[1], Sae Nishimura[1], Yoshio Kunihiro[1], Masahiko Sisido[1] & Kazunori Watanabe[1]

The possibility of spatiotemporally photocontrolling translation holds considerable promise for studies on the biological roles of local translation in cells and tissues. Here we report caged aminoacyl-tRNAs (aa-tRNAs) synthesized using a (7-diethylaminocoumarin-4-yl)methoxycarbonyl (DEACM)-cage compound. DEACM-caged aa-tRNA does not spontaneously deacylate for at least 4 h in neutral aqueous solution, and does not bind to the elongation factor Tu. On irradiation at $\sim 405\,nm$ at $125\,mW\,cm^{-2}$, DEACM-aa-tRNA is converted into active aa-tRNA with a half-life of 19 s. Notably, this rapid uncaging induced by visible light does not impair the translation system. Translation is photoinduced when DEACM-aa-tRNA carrying a CCCG or a CUA anticodon is uncaged in the presence of mRNAs harbouring a CGGG four-base codon or a UAG amber codon, respectively. Protein synthesis is phototriggered in several model systems, including an *in vitro* translation system, an agarose gel, in liposomes and in mammalian cells.

[1] Department of Biomedical Engineering, Okayama University, 3-1-1 Tsushimanaka, Okayama 700-8530, Japan. Correspondence and requests for materials should be addressed to T.O. (email: ohtsuk@okayama-u.ac.jp).

In multicellular organisms, the up- and downregulation of specific proteins in specific cells are related to local biological events, such as development and cell differentiation. Furthermore, the asymmetric distribution of a specific protein or its mRNA in the cytoplasm is related to cellular functions[1]. For example, the localization of *oskar* mRNA to the posterior of the *Drosophila* egg provides spatial determinants for defining the anterior–posterior axis[2], and local protein synthesis in neurons is critical for neuronal function[3,4]. Thus, naturally occurring spatiotemporal control of protein biosynthesis is crucial for cellular functions and early development. Conversely, cellular functions and developmental mechanisms can be revealed by using artificial methods to spatiotemporally control translation.

For controlling translation, methods involving the use of light are effective because light is one of the most easily manipulated external factors. Photocontrol of translation has been achieved by controlling the activity or concentration of an mRNA by using a caged mRNA[5], a photoresponsive 5′-cap[6], caged siRNAs[7–9] or photosensitive RNA carriers[10,11]. Furthermore, translation has been photocontrolled by controlling the activity of an aminoacyl-tRNA (aa-tRNA) by using a caged aa-tRNA[12]. The caged aa-tRNA bearing a four-base anticodon was used to phototrigger the translation of a specific mRNA containing the corresponding four-base codon; however, this caged aa-tRNA containing a photocleavable nitroveratryloxycarbonyl (NVOC) group required ultraviolet irradiation for ∼30 min for its uncaging, which resulted in a reduction in translation efficiency[12]. Because of the ultraviolet damage caused to the translation system and probably to other intracellular systems, the NVOC-caged aa-tRNA could not be readily used to phototrigger translation in a living cell.

In this study, we synthesized a (7-diethylaminocoumarin-4-yl)methoxycarbonyl (DEACM)-caged aa-tRNA (Fig. 1a). DEACM-caged compounds can be uncaged rapidly by irradiation with visible light at ∼400–430 nm (refs 13,14). Thus, as compared with the process used for uncaging NVOC-aa-tRNA, the DEACM-aa-tRNA uncaging process causes lesser photodamage to the translation machinery and to cells. For a caged aa-tRNA to serve as an efficient phototrigger for *in vitro* and *in vivo* translation, it must possess these properties: (1) before irradiation, the caged aa-tRNA must not deacylate in aqueous solutions; (2) before irradiation, it must not participate in translation due to rejection from either elongation factor (EF)-Tu or the ribosome; (3) it must be uncaged rapidly without any damage to the translation system; and (4) after irradiation, it must decode a specific four-base or nonsense codon. Here DEACM-aa-tRNAs containing natural or non-natural aminoacyl moieties (DEACM-Ser-tRNAs and propargylglycyl (PPG)-tRNAs) were prepared and were used for decoding amber and four-base codons. Furthermore, the properties of DEACM-aa-tRNA, including deacylation, EF-Tu binding and photo-uncaging rate were investigated. By using the DEACM-aa-tRNAs, we demonstrated photoinduced *in vitro* translation, photopatterning of protein synthesis in a gelated reaction mixture, photoinduced local translation in liposomes and single-cell-specific translation in a mammalian cell.

## Results

**Photolysis of DEACM-PPG-pdCpA.** A buffered solution (pH 7.6) containing DEACM-PPG-pdCpA was irradiated at ∼405 nm at 125 mW cm$^{-2}$ (75.8% of the light reached the sample in a microtube through the tube wall) at room temperature for 0–30 s, following which the photogenerated DEACM-OH was removed from the irradiated solution, and the fluorescence of the remaining DEACM-PPG-pdCpA was measured (Fig. 1b,c). The fluorescence intensity decreased rapidly with increasing irradiation duration. The decrease in DEACM-PPG-pdCpA followed first-order kinetics with a half-life of 19 s. The photolysis quantum yield was 0.0037, which is similar to that of DEACM-caged galactose derivative (0.0058)[15].

**Stability of DEACM-Ser-tRNA against deacylation.** Previous studies have shown that aa-tRNAs are not stable and are spontaneously deacylated under physiological conditions. For example, the half-lives of Ser-tRNA and Cys-tRNA at pH 7.5 at 30 °C are ∼30 and 20 min, respectively[16,17]. To investigate the stability

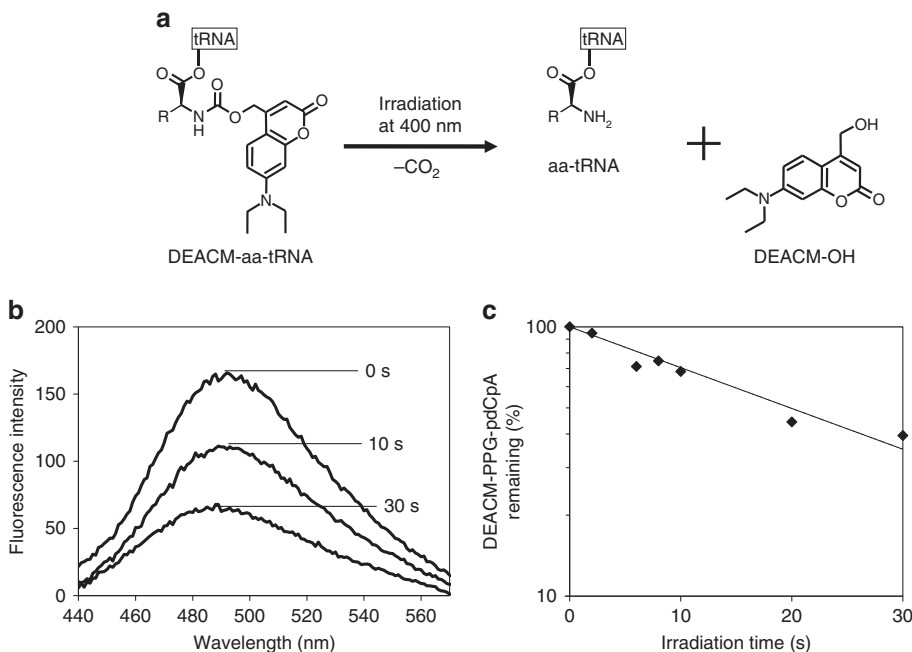

**Figure 1 | Photolysis of DEACM-caged aa-tRNA. (a)** Scheme of DEACM-aa-tRNA photolysis. **(b)** Fluorescence spectra of DEACM-PPG-pdCpA remaining after photoirradiation for 0–30 s. **(c)** Time course of the photolysis of DEACM-PPG-pdCpA. Note that the *y* axis is on a logarithmic scale.

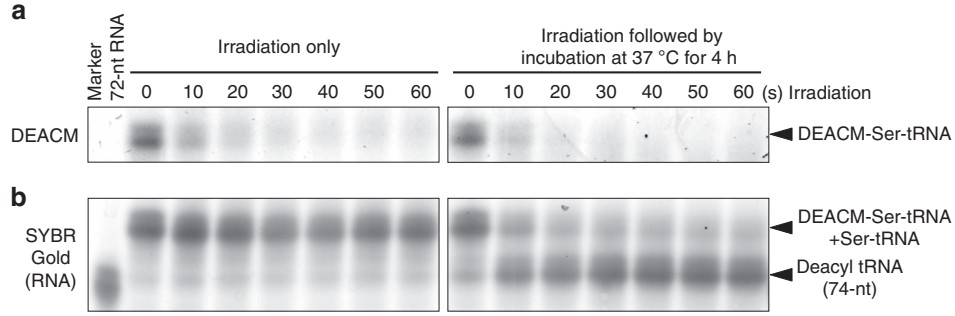

**Figure 2 | Acid-PAGE analysis of photolysis and deacylation of DEACM-Ser-tRNA.** Acid-PAGE analysis was conducted after DEACM-Ser-tRNA samples were irradiated for 0–60 s and then either incubated at 37 °C for 4 h or not incubated post irradiation. (**a**) DEACM fluorescence image. (**b**) Fluorescence image of the corresponding SYBR Gold-stained gel. nt, nucleotide.

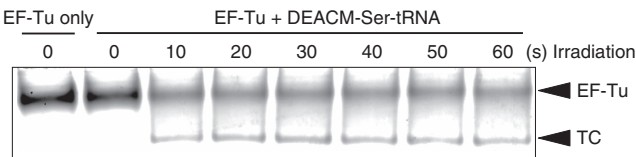

**Figure 3 | Gel-shift analysis of the binding of E. coli EF-Tu to photolysed DEACM-Ser-tRNAs.** Samples were irradiated for the indicated periods, and proteins (EF-Tu) were visualized using SYPRO Red gel stain. TC indicates the ternary complex of EF-Tu, GTP, and aa-tRNA.

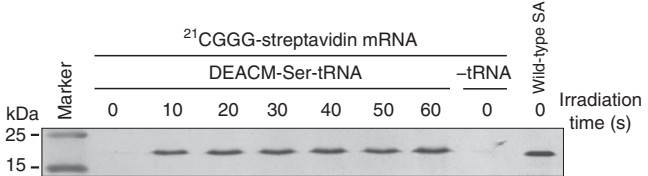

**Figure 4 | Frameshift suppression of the four-base codon CGGG triggered by photoirradiation of DEACM-Ser-tRNA_{CCCG} in *in vitro* translation.** Synthesized proteins were detected by western blotting with an anti-T7-tag antibody.

of the DEACM-protected aa-tRNA, acid-polyacrylamide gel electrophoresis (PAGE) analysis was conducted after samples were irradiated at ~405 nm for 0–60 s and then incubated at 37 °C for 4 h. Photoirradiation induced rapid disappearance of DEACM-Ser-tRNA (Fig. 2a, left), which indicated that DEACM-Ser-tRNA was rapidly photolysed to Ser-tRNA. More than half of the DEACM-Ser-tRNA was photolysed within 20 s by the 405 nm light applied at 184 mW cm$^{-2}$ (Fig. 2a), which was in agreement with the result of the experiment conducted using DEACM-PPG-pdCpA (Fig. 1). The Ser-tRNA generated from the photolysis of DEACM-Ser-tRNA was deacylated at 37 °C (see lanes for 10–60 s irradiation in Fig. 2b, right). By contrast, DEACM-Ser-tRNA (lane for 0 s irradiation in Fig. 2b, right) was not deacylated at 37 °C for at least 4 h. This result indicated that protection with the DEACM group prevents the spontaneous deacylation of aa-tRNAs.

**Binding of DEACM-Ser-tRNA to EF-Tu.** An aa-tRNA binds to EF-Tu together with GTP to form a ternary complex. This process is crucial for the delivery of the amino acid into the ribosomal A-site and for successful incorporation of the amino acid into proteins. Figure 3 shows the results of gel-shift analysis conducted to investigate the ability of DEACM-Ser-tRNA to bind to *Escherichia coli* EF-Tu in the presence of GTP: the ternary complex was not detected before photoirradiation, but appeared after irradiation for 10–20 s; longer irradiation did not appear to cause an increase in the ternary complex. These data indicated that whereas DEACM-Ser-tRNA does not bind to EF-Tu, the photolysis product, Ser-tRNA, forms a ternary complex with EF-Tu. This result confirmed that DEACM-caged aa-tRNA cannot be delivered by EF-Tu to the ribosomal A-site.

**Photoinduced four-base-codon decoding in *in vitro* translation.** *In vitro* translation was phototriggered through the decoding of a

CGGG four-base codon by using DEACM-Ser-tRNA_{CCCG} and streptavidin mRNA-21_{CGGG} (Fig. 4). The CGGG codon was used because this codon is the most efficiently translated in the *E. coli in vitro* translation condition used in the current study[18,19], although the amber codon or other four-base codons may be better suited in other translation conditions. Streptavidin was not detected in the absence of irradiation, but was successfully produced when the reaction mixture was irradiated for >10 s. By comparing the band intensity of the product with that of the marker protein (wild-type streptavidin), the protein yield after 60 s irradiation was estimated to be 1.1 μg per 10 μl translation using streptavidin mRNA-21_{CGGG} and DEACM-Ser-tRNA_{CCCG}. Notably, the yield of wild-type streptavidin was not substantially decreased following irradiation with 405 nm light for up to 2 min (Supplementary Fig. 1), which is longer than the irradiation time sufficient for phototriggering translation using DEACM-aa-tRNA. This is distinct from the translation obtained using NVOC-aa-tRNA[12]: the translation yield was not substantially affected by 50- and 75%-uncaging procedures for DEACM-aa-tRNA, but was significantly decreased by 50- and 75%-uncaging procedures for NVOC-aa-tRNA (Supplementary Fig. 2).

**Photopatterning of protein synthesis in a gel.** Enhanced green fluorescent protein (EGFP) synthesis was photopatterned using EGFP mRNA-214_{CGGG} and DEACM-Ser-tRNA. To minimize the diffusion of the synthesized proteins, the translation reaction mixture was gelled. First, we investigated whether translation was inhibited by gel materials: translation was abolished by 0.5% gelatin gel and 2% acrylamide gel, but not by 0.5–1% agarose gel (Supplementary Fig. 3). Thus, the reaction mixture was gelled with 0.5% agarose. The reaction mixture in the gel on a glass plate was irradiated at 405 nm for 10 s through a photomask (Fig. 5, right) and then incubated at 37 °C for 2 h for the translation reaction. Although EGFP diffusion from the irradiated area

was observed, strong EGFP fluorescence was observed only in the irradiated area (Fig. 5, left). This result indicated that the CGGG codon was decoded by the Ser-tRNA$_{CCCG}$ that was photogenerated from DEACM-Ser-tRNA$_{CCCG}$ in the irradiated area.

**Photoinduced protein synthesis in liposomes**. The translation reaction mixtures containing the *E. coli* PURE system, DEACM-Ser-tRNA$_{CCCG}$ and EGFP mRNA-$30_{CGGG}$ were mixed with lipid films, and liposomes were immediately prepared by vortexing the mixtures for 5 s; the reaction mixtures together with the liposomes were then irradiated for 10 s and incubated for 2 h (Fig. 6a). Under the microscope, EGFP fluorescence was clearly detected only in the irradiated liposomes (Fig. 6b). Because the solution outside the liposomes was also the translation reaction mixture, EGFP must also have been synthesized outside the liposomes in the irradiated area. However, EGFP fluorescence was not clearly detected outside the liposomes even within the irradiated area; thus, EGFP molecules synthesized outside the liposomes likely had diffused away from the irradiated area. In addition, photoinduced protein synthesis in a liposome by using laser light was successfully demonstrated (Supplementary Fig. 4).

**Laser-induced protein synthesis in single mammalian cells**. Finally, we attempted to phototrigger DsRed synthesis through amber suppression in CHO-DsRed-$131_{amber}$ cells (Chinese hamster ovary (CHO) cells stably expressing DsRed mRNA-$131_{amber}$). The cells were microinjected with DEACM-PPG-tRNA$_{CUA}$ and an injection marker (Alexa Fluor 488 C5 maleimide), and then laser-irradiated and incubated at 37 °C. DsRed fluorescence was detected in the irradiated cells after incubation for 5 h and increased in intensity after 8 h incubation (Fig. 7). By contrast, DsRed fluorescence was not detected in unirradiated cells. These results indicated that amber suppression on DsRed mRNA-$131_{amber}$ was successfully phototriggered by laser light in a mammalian cell. Similar results were obtained using CHO cells microinjected with DEACM-PPG-tRNA$_{CUA}$ and an expression vector for mOrange2 mRNA-$131_{amber}$ (Supplementary Fig. 5). The light exposure used in this experiment (3.75 J cm$^{-2}$ at 405 nm) did not damage the CHO cells (Supplementary Fig. 6).

**Discussion**
This study provides a strategy for photoinduced protein synthesis by using DEACM-caged aa-tRNA *in vitro* and in cell-sized liposomes and living cells. DEACM-aa-tRNA cannot act in the

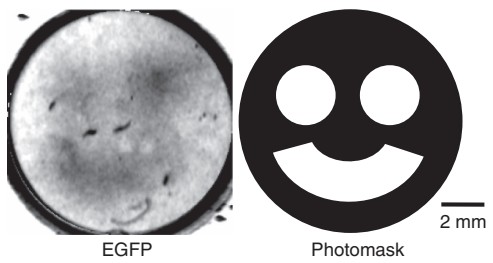

EGFP Photomask

2 mm

**Figure 5 | In-gel synthesis of EGFP by using DEACM-Ser-tRNA and EGFP mRNA-$^{214}$CGGG.** The translation mixture was gelled with 0.5% agarose, and protein synthesis was triggered by irradiating the samples through the photomask (illustrated on the right). The left image shows EGFP fluorescence.

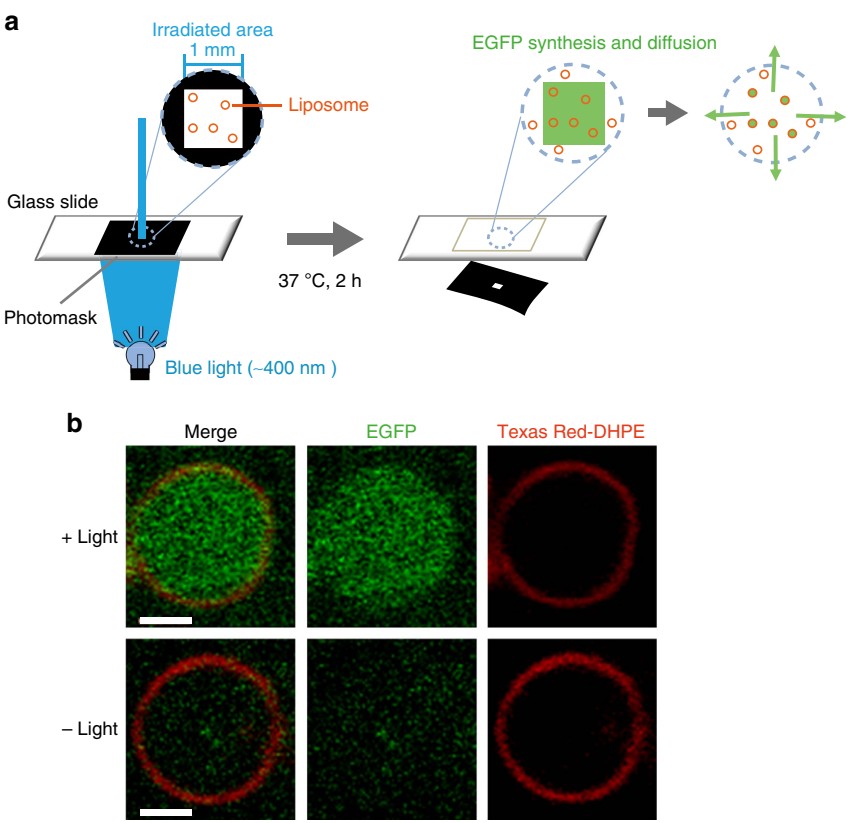

**Figure 6 | Photoinduced EGFP synthesis in liposomes.** (**a**) Method used for irradiating and incubating the translation reaction mixture with liposomes. (**b**) EGFP images of irradiated and unirradiated liposomes. Liposome membranes were visualized using Texas Red-DHPE. Scale bar, 2 μm.

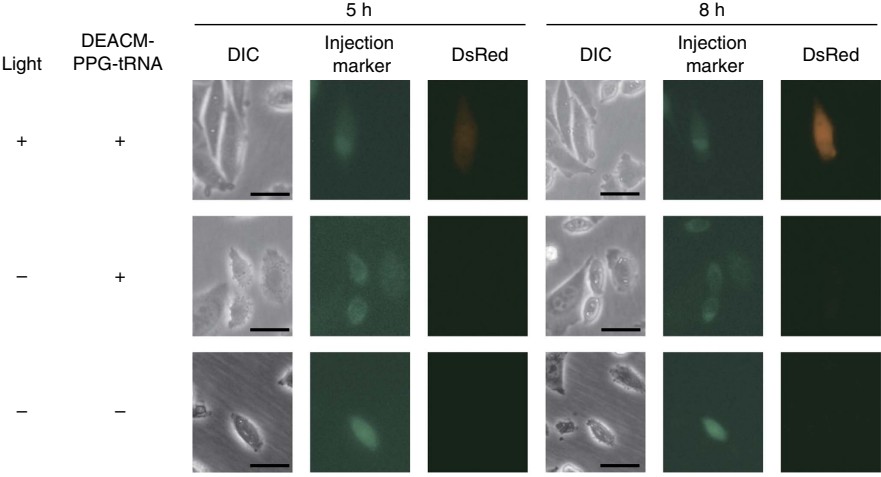

**Figure 7 | Laser-induced translation by using DEACM-PPG-tRNA$_{CUA}$ in individual CHO-DsRed-131$_{amber}$ cells.** The cells were microinjected with a solution containing DEACM-PPG-tRNA$_{CUA}$ and the injection marker Alexa Fluor 488 C5 maleimide (top and middle images), or a solution containing the marker alone (bottom images), irradiated with laser light and incubated for 5 or 8 h. Scale bar, 25 μm.

translation system without irradiation because its amino terminus is protected with the DEACM group, and it cannot be used for peptide bond formation in the ribosomal A-site. DEACM-aa-tRNA did not bind to EF-Tu, and thus it is expected not to enter the ribosomal A-site. The reproducibility of the results (Figs 1–7) was confirmed more than twice.

DEACM-aa-tRNA was rapidly uncaged by visible light at ∼400 nm. This is a notable advantage offered by this caged compound: most other caged compounds require comparatively longer irradiation and/or shorter wavelengths (in the ultraviolet region) for uncaging. In our previous study conducted using NVOC-aa-tRNA, in vitro translation was suppressed by the ultraviolet irradiation that was sufficient for uncaging of the compound[12]. By contrast, the irradiation adequate for uncaging DEACM-aa-tRNA did not reduce translation efficiency.

The method using DEACM-aa-tRNA and visible light allowed spatially controlling translation in liposomes and cells in a highly specific manner. In contrast, in agarose gel, diffusion of the uncaged aa-tRNA and/or the product EGFP protein hindered the clear patterning of translation. Diffusion of the product could be partially inhibited by synthesizing a larger protein. The use of other gelling agents may also contribute to sharper patterns. In the current study, a gel thickness of ∼1 mm was used; however, this method likely can be applied to thicker gels and at deeper levels because 91% of the 405 nm light penetrated through a 10 mm 0.5% agarose gel (Supplementary Fig. 7).

In the mammalian cell experiments, phototriggered DsRed synthesis through amber suppression was successfully achieved by using DEACM-PPG-tRNA$_{CUA}$. DsRed fluorescence was observed in a cell at 5 h after irradiation, and stronger fluorescence was observed at 8 h after irradiation. This time lag may be caused by (1) the translation process, (2) the recovery from the damage from the microinjection and (3) the maturation time of DsRed. Although we used DsRed express, a rapidly maturing variant of DsRed (maturation half-times of DsRed express and wild-type DsRed are 0.7 and 11 h, respectively)[20], mutation with the non-natural amino acid might affect its maturation time. Phototriggered protein synthesis in mammalian cells is a promising strategy for analysing protein-mediated cellular functions. For example, cell cycle-related phenomena could be analysed on the basis of temporally controlled phototriggered protein synthesis, although the time lag between irradiation and protein expression should be taken into account.

Cellular events related to intercellular interactions might be revealed through spatially controlled phototriggered protein synthesis.

Local protein synthesis in cells or tissues is critical for various biological events, and the biological roles of local translation in neurons and eggs in particular have been well studied[1,3]. DEACM-aa-tRNA can be used for phototriggering local translation as demonstrated here by using the gelated translation reaction mixture, liposomes and mammalian cells. On laser irradiation, phototriggered translation was achieved in a liposome (diameter: ∼6 μm) (Supplementary Fig. 4), indicating that the irradiation exposure area can be controlled down to the scale necessary to turn on translation in different parts of the cell. This method of spatiotemporally photocontrolling translation offers a promising approach for investigating the relationship between local translation and biological functions. Therefore, DEACM-aa-tRNA could potentially become a new tool for emerging optogenetic methods[21]. Furthermore, DEACM-aa-tRNA can be used to add specific functions to photogenerated proteins because it can be used as a tool for introducing non-natural amino acids into proteins at specific positions.

## Methods

**Syntheses of DEACM-aa-tRNAs.** DEACM-aa-tRNAs were prepared by means of the chemical acylation method[22] using a DEACM-aa-pdCpA unit and a tRNA lacking the 3′-CA sequence. DEACM-Ser-pdCpA and DEACM-PPG-pdCpA were synthesized as described in Supplementary Information (Supplementary Fig. 8).

A mutant *Methanosarcina barkeri* tRNA$^{Pyl}$, which includes a CCCG four-base anticodon instead of the original GAA anticodon and lacks the 3′-CA sequence[19], was transcribed as follows. To generate DNA templates for transcription, primer extension was performed using 1 μM of each of the primers (5′-CCGGGTAATA CGACTCACTATAGGAAACCTGATCATGTAGATCGAATGGACTC-3′ and 5′-GCGGGAAACCCCGGGAATCTAACCGGCTGAACGGATTCGGGAGT CCATTCGATCTACATG-3′) in a 100 μl reaction mixture containing 0.2 mM dNTPs, 25 units KOD Dash DNA polymerase (Toyobo, Japan) and 10 μl of ×10 buffer #1, with the following temperature programme: 94 °C for 60 s; and three cycles of 94 °C for 30 s, 55 °C for 2 s and 72 °C for 30 s. The primers were designed to complement each other at the 3′-region (∼20 nucleotides). The resultant dsDNA was collected by precipitation with 2-propanol. The transcription reaction was carried out at 37 °C for 4 h in a reaction mixture containing 40 mM Tris-HCl (pH 8.0), 24 mM MgCl$_2$, 5 mM dithiothreitol (DTT), 2 mM spermidine, 0.01% Triton X-100, 10 mM GMP, 2 mM each of NTPs, 1.8 units per ml pyrophosphatase (SIGMA), 750 units per ml T7 RNA polymerase (Takara, Japan) and 200 nM DNA template. The tRNA transcripts were purified in a 10% denaturing polyacrylamide gel. DEACM-Ser-tRNA$_{CCCG}$ and DEACM-PPG-tRNA$_{CCCG}$ were prepared by ligating the transcribed tRNA$_{CCCG}$(-CA) with DEACM-Ser-pdCpA and DEACM-PPG-pdCpA, respectively. The ligation reaction was carried out at 4 °C for 2 h in a

reaction mixture, including 2.75 mM HEPES-NaOH (pH 7.5), 75 mM MgCl$_2$, 16.5 mM DTT, 5 mM ATP, 0.002% BSA, 20 µM tRNA$_{CCCG}$(-CA), 40 µM DEACM-aa-pdCpA and 1.5 units per µl T7 RNA ligase (Takara). After the reaction, the ligated RNA (DEACM-aa-tRNA) was purified by phenol extraction followed by ethanol precipitation. DEACM-PPG-tRNA$_{CUA}$ was prepared by ligating the human amber suppressor tRNA[23] lacking the 3′-CA sequence with DEACM-PPG-pdCpA.

### Photodeprotection of DEACM-Ser-tRNA and acid-PAGE analysis.

The synthesized DEACM-Ser-tRNA was deprotected by irradiating with ~405 nm light (at 184 mW cm$^{-2}$) using an Hg–Xe lamp equipped with a long-pass filter ($\lambda > 345$ nm) at room temperature. The resulting mixture was analysed by acid-PAGE: 4 µl aliquots of 10 µM DEACM-Ser-tRNA in 20 mM HEPES-KOH (pH 7.6) were irradiated for different periods, and then the samples were incubated at 37 °C. The solutions were mixed with equal volumes of gel-loading solution containing 200 mM NaOAc (pH 5.0), 20 mM EDTA and 7 M urea, and then electrophoresed at 4 °C for 18 h under 500 V in a 7.5% polyacrylamide gel in a buffer containing 100 mM NaOAc (pH 5.0) and 10 mM EDTA. The gel was stained with SYBR Gold (Invitrogen, USA) and fluorescence images of the gels were acquired using an FMBIO III-SC01 instrument (Hitachi, Japan) (imaging: SYBR Gold: $\lambda_{ex} = 488$ nm and $\lambda_{em} = 545$–565 nm; DEACM group: $\lambda_{ex} = 405$ nm and $\lambda_{em} = 450$–470 nm).

### Measurement of photolysis rate.

DEACM-PPG-pdCpA (75 pmol) dissolved in 50 µl of 1 mM HEPES-KOH (pH 7.5) was irradiated using the Hg–Xe lamp (at ~405 nm, 125 mW cm$^{-2}$). Photogenerated DEACM-OH was removed from the irradiated solution by using diethyl ether, and the aqueous layer containing the remaining DEACM-PPG-pdCpA was collected, dried in vacuo and redissolved in 1 mM HEPES-KOH (pH 7.5). Sample fluorescence was measured using an FP-6600 spectrofluorometer (Jasco, Japan) ($\lambda_{ex} = 400$ nm), and collected data were normalized relative to 1 mM HEPES-KOH (pH 7.5) without DEACM-PPG-pdCpA. The half-life and the quantum yield of DEACM-PPG-pdCpA were calculated from its fluorescence intensity.

### Gel-shift analysis using EF-Tu and DEACM-aa-tRNAs.

E. coli EF-Tu was prepared as follows[24]. BL21 (DE3) E. coli cells were transformed with the EF-Tu expression vector[25] for the production of His-tagged EF-Tu, which was purified by Ni-NTA agarose (Qiagen) column chromatography. The purified EF-Tu solution was dialysed against a buffer containing 50 mM HEPES-KOH (pH 7.6), 50 mM KCl, 1 mM DTT, 2 µM GDP and 10% glycerol.

DEACM-Ser-tRNA binding to E. coli EF-Tu was examined by gel-shift analysis. A ternary complex of EF-Tu, GTP and aa-tRNA was prepared as follows. EF-Tu (10 µM) was preincubated with GTP (1 mM) at 37 °C for 15 min in a 5 µl total volume containing 70 mM HEPES-KOH (pH 7.6), 52 mM NH$_4$OAc, 8 mM Mg(OAc)$_2$, 30 mM KCl, 0.8 mM DTT, 1.6 µM GDP, 6% glycerol, 10 mM phosphoenolpyruvate and 0.08 unit per µl pyruvate kinase. To the preincubated EF-Tu solution, 3 µl of 3.3 µM DEACM-aa-tRNA (dissolved in 6 mM KOAc) and 2 µl of ternary complex buffer, containing 150 mM HEPES-KOH (pH 7.6), 195 mM NH$_4$OAc and 30 mM Mg(OAc)$_2$,were added. The mixture was incubated at 37 °C for 10 min. Electrophoresis of the ternary complex mixtures (5 µl), including EF-Tu (5 µM) and DEACM-Ser-tRNA (1 µM) was performed using 8 % polyacrylamide gels at 4 °C in a buffer containing 50 mM Tris-HCl (pH 6.8), 65 mM NH$_4$OAc and 10 mM Mg(OAc)$_2$. Gels were stained with SYPRO Red (Takara, Japan). SYPRO Red imaging was carried out using an FLA-9000 (Fujifilm, Japan) ($\lambda_{ex} = 532$ nm and $\lambda_{em} > 575$ nm).

### In vitro synthesis of streptavidin in an E. coli S30 system.

Streptavidin mRNA harbouring a CGGG codon at amino-acid position 21 (streptavidin mRNA-$^{21}_{CGGG}$) was prepared by in vitro transcription. The transcription reaction was performed at 37 °C for 6 h in a reaction mixture containing 40 mM Tris-HCl (pH 8.0), 20 mM MgCl$_2$, 5 mM DTT, 2 mM spermidine, 4 mM of each NTP, 50 units per ml pyrophosphatase, 400 units per ml RNase inhibitor, 2,000 units per ml T7 RNA polymerase and 10 µg per ml template DNA, To the transcript, one volume of prechilled 5 M ammonium acetate was added, and the solution was placed on ice for 20 min. The mRNA was collected by centrifugation at 24,000 g at 4 °C for 20 min. The mRNA was purified by phenol/chloroform extraction and ethanol precipitation. A T7-tag sequence was encoded at the N terminus of the streptavidin mRNA to allow protein detection with an anti-T7 antibody. To decode the CGGG four-base codon, the aforementioned DEACM-Ser-tRNA$_{CCCG}$ was used. In vitro translation and western blotting were performed as follows. A 10 µl in vitro translation mixture was prepared containing 2 µl of E. coli S30 Extract for Linear Templates (Promega), 55 mM HEPES-KOH (pH 7.5), 210 mM potassium glutamate, 6.9 mM ammonium acetate, 1.7 mM DTT, 1.2 mM ATP, 0.28 mM GTP, 26 mM phosphoenolpyruvate, 1 mM spermidine, 1.9% polyethyleneglycol-8000, 35 µg ml$^{-1}$ folinic acid, 12 mM magnesium acetate, 0.1 mM of each amino acid, 8 µg of mRNA and 0.1 nmol of DEACM-Ser- tRNA$_{CCCG}$. The reaction mixture was irradiated using the Hg–Xe lamp (at ~405 nm, 184 mW cm$^{-2}$), incubated at 37 °C for 1 h and then separated using a 15% SDS–polyacrylamide gel. Proteins were transferred to a polyvinylidene difluoride membrane (Bio-Rad) and western blotted

using an anti-T7-tag monoclonal antibody (Novagen) and the ProtoBlot II AP system (Promega). The uncropped scan of the blot (Fig. 4) is shown in Supplementary Fig. 9.

### Photopatterning of EGFP synthesis in agarose gels.

EGFP mRNA carrying a CGGG codon at amino-acid position 214 (EGFP mRNA-$^{214}_{CGGG}$) was prepared as described above. The translation mixture including E. coli S30 extract, EGFP mRNA-$^{214}_{CGGG}$ and DEACM-Ser-tRNA$_{CCCG}$ was the same as the mixture described for streptavidin synthesis, except that 0.5% Seaplaque agarose (Takara) was included. The reaction mixture (80 µl) at 37 °C was loaded and spread on the bottom of a glass-bottom dish (glass diameter 10 mm), and then gelled by cooling on ice for 1 min. The gel was immediately irradiated through a photomask for 10 s at ~405 nm at 184 mW cm$^{-2}$. The dish including the reaction mixture was placed in an incubator at 37 °C for 2 h in a closed box that was humidified to prevent drying of the reaction mixture. After incubation, the dish containing the reaction mixture was analysed on the Hitachi FMBIO III-SC01 fluorescence image analyser ($\lambda_{ex} = 488$ nm and $\lambda_{em} = 505$–545 nm).

### Photoinduced EGFP synthesis in liposomes.

A 50 µl methanol/chloroform (1:2 v/v) solution containing 1 mM 1,2-dioleoyl-sn-glycero-3-phosphocholine (DOPC; Fluka) and 1 µM Texas Red-1,2-dihexadecanoyl-sn-glycero-3-phosphoethanolamine (Texas Red-DHPE; Invitrogen) was dried under N$_2$ gas in a glass test tube and left to dry overnight under vacuum to obtain lamellar lipid films. The films were hydrated with 5 µl of cold (~4 °C) cell-free translation reaction mixture by vortexing for 5 s immediately after adding the mixture. The 5 µl translation reaction mixture contained 2 µl of solution A, 1.5 µl of solution B, 0.5 µg of EGFP mRNA-$^{30}_{CGGG}$ and 4 µg of DEACM-Ser-tRNA$_{CCCG}$. Solutions A and B were included in the PURExpress kit (New England BioLabs) that is based on the PURE system[26]. Immediately after completion of the lipid-film hydration procedure, which generates giant lipid vesicles (5–15 nm diameter), a 2.5 µl suspension of the vesicles was placed on a glass slide and sealed with a cover glass (0.12–0.17 mm); the cover glass was fixed in the centre of the glass slide with double-sided adhesive tape (~0.2 mm thick). Immediately after the sample on the glass slide was sealed, the sample was irradiated through a photomask featuring a 1 × 1 mm square window by using the Hg–Xe lamp (125 mW cm$^{-2}$ at ~405 nm) (Fig. 6a). Immediately after the irradiation, the sample on the glass slide was incubated at 37 °C for 2 h to allow the translation reaction to occur, and then examined under a confocal laser-scanning microscope (FLUOVIEW FV-1000, Olympus, Japan) (imaging: Texas Red: $\lambda_{ex} = 543$ nm and $\lambda_{em} = 555$–655 nm; EGFP: $\lambda_{ex} = 488$ nm and $\lambda_{em} = 500$–600 nm).

### Photo-dependent amber suppression in mammalian cells.

The DsRed expression vector including an amber codon at codon position 131 (pcDNA5-DsRed-$^{131}_{amber}$) was prepared by subcloning the DsRed gene from pDsRed-Express-N1 (Clontech) into pcDNA5/FRT/TO Flp-In expression vector (Invitrogen) between BamHI and XhoI sites, and then mutating the codon position 131 by using a QuikChange site-directed mutagenesis kit (Stratagene). CHO cells stably expressing DsRed mRNA-$^{131}_{amber}$ (CHO-DsRed-$^{131}_{amber}$) were prepared by transfecting the Flp-In CHO cell line (Invitrogen) with pcDNA5-DsRed-$^{131}_{amber}$ according to the standard protocol of the Flp-In system. The CHO-DsRed-$^{131}_{amber}$ cells were grown at 37 °C and 5% CO$_2$ in Ham's F12 medium supplemented with 10% fetal bovine serum (Sigma) and 1% streptomycin/penicillin (Gibco).

CHO-DsRed-$^{131}_{amber}$ cells plated in a 35 mm dish (~70% confluence, in Ham's F12 without antibiotics) were microinjected with 1 × PBS solution including 36 µM DEACM-PPG-tRNA$_{CUA}$ and, as an injection marker, 10 µM Alexa Fluor 488 C5 maleimide (Molecular Probes). Microinjections were performed using Eppendorf Femtojet microinjection equipment and Femtotip microinjection capillary tips at 100–150 hPa. The cells were irradiated for 30 s by using the Hg–Xe lamp (~405 nm, 125 mW cm$^{-2}$) and then incubated for 4 h at 37 °C and 5% CO$_2$. Cellular fluorescence was imaged using a fluorescence microscope (Olympus IX51/IX2-FL-1/MP5Mc/OL-2).

### Data availability.

The data that support the findings of this study are available from the corresponding author on request.

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

## Acknowledgements

We thank Prof. Akira Sakakura (Okayama University) and Dr Ichiro Hayakawa (Okayama University) for help with the organic synthesis, Prof. Toshiaki Furuta (Toho University, Japan) for help with calculating the quantum yield and Yuta Fujiwara (Okayama University) for help with the preparation of caged aa-tRNA. This work was funded by the JSPS KAKENHI (No. 25282232) and Grants-in-Aid for Scientific Research on Innovative Areas 'Nanomedicine Molecular Science' to T.O. (26107711).

## Author contributions

T.O., K.W. and M.S. designed the study; T.O. analysed the data and wrote the paper; K.W. partially wrote the paper; S.K. carried out the photolysis and cell experiments; S.N. conducted the liposome experiments; Y.K. conducted the *in vitro* translation experiments. All authors have read and approved the manuscript before submission.

## Additional information

**Competing financial interests:** The authors declare no competing financial interests.

