## [Peer Review File · Nature Communications]

Reviewer #1 (Remarks to the Author)

Summary

This article by Ohtsuki et al. describes the photo-triggered control of protein synthesis. The key idea is that the authors use a chemical agent, DEACM, to cage an aminoacyl-tRNA (aa-tRNA). In the absence of visible light around 400-430nm, the caged-aa-tRNA remains stable and is not a substrate for the translation apparatus. However, in the presence of light, the caged-substrate undergoes photolysis to make available the aa-tRNA substrate. This work builds directly off and improves upon the authors' previous work (ref 12), which had been limited because the UV irradiation necessary for a different cage (NVOC) was deleterious to biological function. The authors demonstrated the ability to use their new system to photoinduce in vitro translation, photopattern protein synthesis, photoinduce in vitro translation in liposomes, and enable translation in mammalian cells. The extension to mammalian cells is potentially exciting, but the data seem early. Broadly, the idea of spatiotemporally photocontrolling translation has applications for synthetic biology, minimal cells, and optogenetics.

In general, this is an intriguing manuscript and a great research area. However, my enthusiasm for publishing in Nature Communications was lessened because my understanding is that the concept of the idea had been previously published by the same group. In addition, the story seems underdeveloped. It felt as though the authors identified a better cage than in previous works and then applied it to as many applications as possible. However, in doing so, the results were mostly qualitative and it seems as though the system is inefficient, which could limit broad interest and applicability. For these reasons, I struggled with potential impact of the work, despite the fact it is an interesting topic.

General remarks/concerns

- What about replicates? It is unclear to me how many replicates have been run for each figure and how reproducible the data are. This needs to be addressed please.
- In Figure 3, it would enhance impact to make the results more quantitative. Also, it seems that the bulk of the product is not ternary complex (TC), why? What is the exact efficiency of uncaging? Or is this an issue with EF:Tu binding? It seems that the authors could optimize and work out conditions to make most product ternary complex. Have the authors tried a positive control with a non-caged aa-tRNA substrate? What is the specific affinity (in μM) between EF-Tu and the aa-tRNA in their hands? and how does this compare to literature values?
- In Figure 4, why did the authors use a 4 base codon? And why the CGGG codon specifically? I believe that the translation data comes from the Promega kit lysate, so I was wondering if they tried other codons; for example the amber codon? It would be better to directly compare in a head-to-head fashion the two different Cages (DEACM vs. NVOC) to ensure the stated improvements are not batch to batch variability, but truly the effect of the cage. How much protein is synthesized (in μM)? This is important to use this approach to generate enough protein for biochemistry experiments.
- In the patterning work, the system seems fairly inefficient. I agree our eyes see a hint of the smiley face, but how can the authors improve the work to sharpen the bands if we envision using this approach for patterning?
- In the liposome work, how does expression compare to a system where wild type GFP is made? It would be good to benchmark the system against this.
- I am trying to understand why the significant time-lag in the mammalian cell experiments. Can the authors clarify this? Was it expected? Also, would the cage substrate be stable this long? Has

this been verified? And how would the spatiotemporal control in this setting would be used? An application of the technology would enhance impact.

- Please double check grammar and spelling - there are some typos.

Reviewer #2 (Remarks to the Author)

The authors present a clear concise description of their work with caged aminoacyl-tRNAs that are liberated for translation upon exposure to 405 nm light. The reported approach is an improvement upon their previously reported work. In this work the authors use a different chemical to cage the tRNA (DEACM) that requires an order of magnitude less 405 nm light exposure time than a previously reported approach (NVOC-caged aminoacyl-tRNA). This is important as in vitro translation is not inhibited by the < 2 min exposure time required; whereas, in vitro translation was inhibited by the longer ~ 30 min exposure time required to liberate NVOC-caged aminoacyl-tRNA. Also the authors mention how extended UV light exposure could damage cells if the technology was to be used in vivo. The paper is well written and concise. I do have the following reservations that if addressed would greatly strengthen the paper.

1) The major motivation for the work presented is spatiotemporal control of protein biosynthesis in cells and tissues. However this doesn't seem plausible with the approach as currently presented. For in vivo work, injection of the caged-tRNA into the cells is required. While this might be reasonable for analyzing a single cell, the authors didn't provide evidence of controlling the irradiation exposure area down to the scale necessary (< 10 microns) to thus enabling a researcher to turn on or off translation in different parts of the cell. The smallest irradiated area shown was ~ 1 mm. This scale might be relevant for tissues, however microinjection of the caged tRNA into all the cells in a tissue sample seems unreasonably difficult.

2) No evidence is provided that the 30 sec UV light exposure used for DEZCM-caged aminoacyl-tRNA liberation in CHO cells did not damage the CHO cells used. As light exposure time is a major motivation for the new caged tRNA system, further experimentation or at least discussion and citations of works which have measure such an effect would be helpful.

3) If using this approach in tissue samples, there is some concern about what intensity of light would be available to deeper levels. The gel data in figure 5 was interesting and perhaps seeing a cross-section at points of exposure could help with realizing the level of tRNA liberation deeper in a gel. I realize there is some diffusion that may hinder clearly assessing this exposure level. However, combining the liposomes in a gel or even in a tissue and then observing the cross-section would help alleviate this concern.

4) The authors did a nice job of showing that EF-Tu is not directly binding DEACM-caged aminoacyl-tRNA and also that detectable protein synthesis with the caged aminoacyl-tRNA did not appear to occur until after irradiation. However, they do not directly measure DEACM-caged aminoacyl-tRNA interaction with the ribosomal A-site. I would thus recommend tempering the language in the concluding sentence of the 1st paragraph of the discussion section. Perhaps replacing "cannot" with "should not" or "likely cannot".

Overall the work is well performed and of interest to the protein translation and unnatural amino acid communities.

Reviewer #3 (Remarks to the Author)

Ohtsuki and coworkers present the design and study of a photoactivatable protein protected by the aminocoumarinyl group. The experiments are well done and the application is interesting.

The photochemistry of diethylaminocoumarinyl moiety has already been established and applied in many studies before. From this point of view, the results seem not to be sufficiently novel, however, my expertise is not strong enough for a proper evaluation of the tRNA application presented in the paper. I have one major and several minor points that should be addressed before the work is published.

Major point

The authors report that the coumarinyl moiety release follows the first-order kinetics, which is expected, but provide only the experimental time, wavelength and irradiance. This may be useful for repeating the experiment but uncaging efficiencies can only be described by the quantum yield.

Minor points

Page 7: Light is not toxic. It may only trigger the formation of photoproducts which are toxic.
Page 7: The Discussion is not very descriptive and does not sufficiently emphasize the novelty of this work.

Figure 1c: The y-axis is a logarithmic function; this should be shown. The release seems to level off at approximately 30 s (60% conversion). Is the full conversion eventually achieved?

Responses to the reviewers' comments

We thank the reviewers for their useful comments, which have greatly helped us improving the manuscript. Please find our point-by-point responses to the comments below. Changes in the revised manuscript are highlighted in red font.

Reviewer #1:

1) What about replicates? It is unclear to me how many replicates have been run for each figure and how reproducible the data are. This needs to be addressed please.

Response: Reproducibility was confirmed three times for experimental data shown in Figures 1, 2, 3, 4 and 6, and twice for those shown in Figures 5 and 7. Reproducibility of the data in Figures 5 and 7 was further confirmed in similar experiments, such as the one shown in Figure S4. We have added a sentence on the reproducibility in the first paragraph of the Discussion (p. 7).

2) In Figure 3, it would enhance impact to make the results more quantitative. Also, it seems that the bulk of the product is not ternary complex (TC), why? What is the exact efficiency of uncaging? Or is this an issue with EF:Tu binding? It seems that the authors could optimize and work out conditions to make most product ternary complex. Have the authors tried a positive control with a non-caged aa-tRNA substrate? What is the specific affinity (in microM) between EF-Tu and the aa-tRNA in their hands? and how does this compare to literature values?

Response: In the experiment shown in Figure 3, the efficiency of uncaging at each time point under the experimental conditions used could be roughly estimated from the following figure (this is not shown in the manuscript, because it is similar to Figure 2a (left)).

Uncaging efficiency after 10 s was more than 70%, and that after 30 s was more than 90%. Most of EF-Tu did not form ternary complex (Fig. 3); however, this was not because of poor uncaging

efficiency but because the concentration of EF-Tu (5 μ M) was higher than that of DEACM-Ser-tRNA (1 μ M). This information was added to “Gel-shift analysis using EF-Tu and DEACM-aa-tRNA” in the Methods (p. 10).

Most of the K_d values for the binding of EF-Tu/GTP and aa-tRNAs were estimated by using the filter assay method using aa-tRNAs enzymatically aminoacylated with RI-labeled amino acids; the K_d values are generally 2–50 nM (LaRiviere F. J. et al., *Science*, 294, pp.165-168, 2001). However, we could not use RI-labeled amino acids because our DEACM-aa-tRNAs were prepared by the chemical method starting from ~1 mmol materials; it would be extremely expensive to use RI-labeled amino acids. In the two weeks, we have tried to estimate the K_d value by the gel-shift assay; however, it was difficult because the band in the gel-shift assay was too diffuse to allow quantification and the recombinant EF-Tu was partially inactive (it seems difficult to obtain 100% active EF-Tu). We used up the remaining DEACM-aa-tRNA and non-caged aa-tRNA.

The difference between uncaged DEACM-aa-tRNA and non-caged aa-tRNA is that uncaged DEACM-aa-tRNA includes the free DEACM group detached from the DEACM-aa-tRNA. The yield of translation of wild-type EGFP mRNA with DEACM-aa-tRNA_{CUA} was largely unaffected by the uncaging procedure to generate free DEACM group (Fig. S1). This result indicates that the free DEACM group does not inhibit the translation machinery, including EF-Tu binding to aa-tRNAs.

3) In Figure 4, why did the authors use a 4 base codon? And why the CGGG codon specifically? I believe that the translation data comes from the Promega kit lysate, so I was wondering if they tried other codons; for example the amber codon? It would be better to directly compare in a head-to-head fashion the two different Cages (DEACM vs. NVOC) to ensure the stated improvements are not batch to batch variability, but truly the effect of the cage. How much protein is synthesized (in μ M)? This is important to use this approach to generate enough protein for biochemistry experiments.

Response: In Figure 4, we used the CGGG codon with *E. coli* S30 Extract (Promega) because CGGG is one of the most efficient four-base codons (CGGG, CGGU, and GGGU) in the *E. coli* *in vitro* translation system (Hohsaka et al., *Biochemistry*, 40, pp. 11060-11064, 2001), and because we had the improved tRNA_{CCCG} (mutant *Methanosarcina barkeri* tRNA^{Pyl}_{CCCG}) that efficiently translates the CGGG codon (Ohtsuki et al., *FEBS Lett.*, 579, pp. 6769-6774, 2005). By comparing the band intensity of the product with that of the marker protein (wild-type streptavidin), the protein yield after 60-s irradiation was estimated to be 1.1 μ g per 10 μ L of translation using streptavidin mRNA-²¹_{CGGG} and DEACM-Ser-tRNA_{CCCG}. In our translation

condition using the *E. coli* S30 lysate, the amber codon is less efficiently translated than the CGGG codon. In our preliminary work, translation reactions using the amber codon were attempted several times but the yield was always less than 70% of the yield using the CGGG codon. We have clarified why we used this particular four-base codon under “Photoinduced 4-base-codon decoding in *in vitro* translation” in the Results (p. 5) as follows: “The CGGG codon was employed because this codon is the most efficiently translated in the *E. coli in vitro* translation condition used in the current study (Hohsaka et al., 2001; Ohtsuki et al., 2005), although the amber codon or other four-base codons may be better suited in other translation conditions” We have added the following sentences about the yield in the Results (p. 5): “By comparing the band intensity of the product with that of the marker protein (wild-type streptavidin), the protein yield after 60-s irradiation was estimated to be 1.1 µg per 10 µL translation using streptavidin mRNA-²¹_{CGGG} and DEACM-Ser-tRNA_{CCCG}.”

We have confirmed that the translation yield was not substantially decreased by 50%- and 75%-uncaging procedures for DEACM-aa-tRNA, but was substantially decreased by 50%- and 75%-uncaging procedures for NVOC-aa-tRNA. The 50%-uncaging procedure is 25-s irradiation at 405 nm for DEACM-aa-tRNA, and 27.7-min irradiation at 365 nm. We have added the following figure (Fig. S2).

Figure S2. Yield of EGFP synthesized in *in vitro* translation using wild-type EGFP mRNA after irradiation. A 10- μ L *in vitro* translation mixture as described in the Methods was irradiated and incubated as indicated in each panel in (a)-(c). The translation reactions were carried out in the presence of 0.1 nmol of DEACM-PPG-tRNA_{CUA} (a) or NVOC-PPG-tRNA_{CUA} (b), or in the absence of caged aa-tRNA (c). The caged PPG-tRNA_{CUA} molecules were unnecessary for the translation of the EGFP mRNA lacking the UAG codon. These molecules were added to investigate their inhibitory effect on translation. The translation products were separated using a 15% SDS-polyacrylamide gel, and EGFP fluorescence was captured using an FLA-9000 imager (Fujifilm, Japan) with an $\lambda_{\text{ex}} = 489 \text{ nm}$ and $\lambda_{\text{em}} = 508 \text{ nm}$. The fluorescence band intensities of EGFP were calculated using ImageJ. N = 3.

4) In the patterning work, the system seems fairly inefficient. I agree our eyes see a hint of the smiley face, but how can the authors improve the work to sharpen the bands if we envision using this approach for patterning?

Response: As correctly indicated by the reviewer, the patterning work was inefficient. However, we consider that it is important to show Figure 5 to demonstrate the limitation of our method in a gel. We have added the following sentences about the patterning work, including how to improve this work, in the third paragraph of the Discussion (p. 7): “The method using DEACM-aa-tRNA and visible light allowed spatially controlling translation in liposomes and cells in a highly specific manner. In contrast, in agarose gel, diffusion of the uncaged aa-tRNA and/or the product EGFP protein hindered the clear patterning of translation. Diffusion of the product could be partially inhibited by synthesizing a larger protein. The use of other gelling agents may also contribute to sharper patterns.”

5) In the liposome work, how does expression compare to a system where wild type GFP is made? It would be good to benchmark the system against this.

Response: To compare the phototriggered EGFP expression to a system where wild-type EGFP is produced, the following five processes are necessary: (1) wild-type EGFP should be synthesized in a liposome, although EGFP is also synthesized outside the liposome, unlike phototriggered protein synthesis, (2) the liposome should be washed with a buffer similar to the translation reaction mixture to remove EGFP proteins synthesized outside the liposome by a centrifugation method to precipitate liposomes, (3) the liposome should be placed on a glass slide and sealed with a cover glass, (4) the liposome on the glass slide should be imaged using a confocal laser-scanning microscope, (5) from the image, the EGFP fluorescence intensity in the liposome should be quantified and compared to the photo-dependently synthesized EGFP in the liposome. Liposomes of similar size should be compared because the larger the liposome, the stronger the fluorescence is, even if the same concentration of EGFP is used.

To allow accurate comparison, EGFP synthesis (step (1)) should be performed on a glass slide sealed with a cover glass because phototriggered EGFP synthesis is carried out on a glass slide. However, liposome washing (step (2)) cannot be performed using a glass slide because it includes only 3 μL of the liposome solution. Step (2) requires a large volume of liposome solution (including expensive PURExpress kit) because the centrifugation induces disruption of liposomes and gentle centrifugation leads to a low rate of liposome collection. Thus, it would be difficult to compare both methods in liposomes.

6) I am trying to understand why the significant time-lag in the mammalian cell experiments. Can the authors clarify this? Was it expected? Also, would the cage substrate be stable this long? Has this been verified? And how would the spatiotemporal control in this setting would be used? An application of the technology would enhance impact.

Response: We have added clarifications for the time lag observed and the applications of the technology in the fourth paragraph of the Discussion (p. 7).

“DsRed fluorescence was observed in a cell at 5 h after irradiation, and stronger fluorescence was observed at 8 h after irradiation. This time lag may be caused by (1) the translation process, (2) the recovery from the damage from the microinjection, and (3) the maturation time of DsRed. Although we used DsRed express, a rapidly maturing variant of DsRed (maturation half-times of DsRed express and wild-type DsRed are 0.7 h and 11 h, respectively)²⁰, mutation with the nonnatural amino acid might affect its maturation time. Phototriggered protein synthesis in mammalian cells is a promising strategy for analyzing protein-mediated cellular functions. For example, cell cycle-related phenomena could be analyzed on the basis of temporally controlled phototriggered protein synthesis, although the time lag between irradiation and protein expression should be taken into account. Cellular events related to intercellular interactions might be revealed through spatially controlled phototriggered protein synthesis.”

7) Please double check grammar and spelling - there are some typos.

Response: We had the grammar and spelling checked by a professional English language editor (Editage; <http://www.editage.jp/>).

Reviewer #2:

1) The major motivation for the work presented is spatiotemporal control of protein biosynthesis in cells and tissues. However this doesn't seem plausible with the approach as currently presented. For in vivo work, injection of the caged-tRNA into the cells is required. While this might be reasonable for analyzing a single cell, the authors didn't provide evidence of controlling the irradiation exposure area down to the scale necessary (< 10 microns) to thus enabling a researcher to turn on or off translation in different parts of the cell. The smallest irradiated area shown was ~ 1 mm. This scale might be relevant for tissues, however microinjection of the caged tRNA into all the cells in a tissue sample seems unreasonably difficult.

Response: In Figure S4, we demonstrated the irradiation of a single liposome by using laser light of a confocal laser-scanning microscope, and translation was successfully photoinduced in the liposome. The diameter of this liposome was about 6 μm . In general, the irradiation

exposure area can be less than 10 μm by using confocal laser-scanning microscopes. We have highlighted this in the fourth paragraph of the Discussion (p. 8): “Upon laser irradiation, phototriggered translation was achieved in a liposome (diameter: $\sim 6\text{-}\mu\text{m}$) (Fig. S4), indicating that the irradiation exposure area can be controlled down to the scale necessary to turn on translation in different parts of the cell.”

2) No evidence is provided that the 30 sec UV light exposure used for DEZCM-caged aminoacyl-tRNA liberation in CHO cells did not damage the CHO cells used. As light exposure time is a major motivation for the new caged tRNA system, further experimentation or at least discussion and citations of works which have measure such an effect would be helpful.

Response: We have newly added Figure S6 on the cytotoxicity of irradiation in the Supporting Information, and the following sentence was added at the end of Results: “The light exposure used in this experiment (3.75 J/cm^2 at 405 nm) did not damage the CHO cells (Fig. S6).”

Figure S6. Cytotoxicity after irradiation. Cytotoxicity was evaluated using CHO cells and a Cytotoxicity Detection Kit (LDH) (Roche, Switzerland). At 24 h after 3.75 J/cm^2 or 13.6 J/cm^2 irradiation at 405 nm (3.75 J/cm^2 is the same intensity as that used in the experiment in Figure 7 [30 s, 125 mW/cm^2]), an aliquot of medium was removed and mixed with the cytotoxicity detection substrate according to the manufacturer’s protocol, and the absorbance at 490 nm was measured. The values were normalized by subtracting the values measured in wells containing no cells, and the values for cells that were incubated with medium containing 0.01% NP-40 were considered to represent 100% cytotoxicity.

3) If using this approach in tissue samples, there is some concern about what intensity of light would be available to deeper levels. The gel data in figure 5 was interesting and perhaps seeing a cross-section at points of exposure could help with realizing the level of tRNA liberation deeper in a gel. I realize there is some diffusion that may hinder clearly assessing this exposure level. However, combining the liposomes in a gel or even in a tissue and then observing the

cross-section would help alleviate this concern.

Response: We have measured the light intensity through a 0.5% agarose gel (the same as the one used in the experiment shown in Figure 5). We have newly added Fig. S7 showing the results and we have discussed the results in the third paragraph of the Discussion: “In the current study, a gel thickness of ~1 mm was used; however, this method likely can be applied to thicker gels and at deeper levels because 91% of the 405-nm light penetrated through a 10-mm 0.5% agarose gel (Fig. S7).”

Figure S7. Light intensity through a 0.5% agarose gel. Light intensity at 405 nm was measured using an ADCMT Optical Power Meter 8230E. The distance between the detector and the light source (a Hg-Xe lamp equipped with a long-pass filter) was constant (55 mm) in each analysis, and the gel was inserted between the detector and the light source.

4) The authors did a nice job of showing that EF-Tu is not directly binding DEACM-caged aminoacyl-tRNA and also that detectable protein synthesis with the caged aminoacyl-tRNA did not appear to occur until after irradiation. However, they do not directly measure DEACM-caged aminoacyl-tRNA interaction with the ribosomal A-site. I would thus recommend tempering the language in the concluding sentence of the 1st paragraph of the discussion section. Perhaps replacing "cannot" with "should not" or "likely cannot".

Rebresponse: In agreement with the reviewer’s comment, we have corrected the indicated sentence as follows: “DEACM-aa-tRNA did not bind to EF-Tu, and thus is expected not to enter the ribosomal A-site.”

Reviewer #3:

1) Major point

The authors report that the coumarinyl moiety release follows the first-order kinetics, which is expected, but provide only the experimental time, wavelength and irradiance. This may be useful for repeating the experiment but uncaging efficiencies can only be described by the quantum yield.

Response: We have calculated the quantum yield (0.0037) and we have added this to the Results section under “Photolysis of DEACM-PPG-pdCpA” (p. 4).

Minor points

2) Page 7: Light is not toxic. It may only trigger the formation of photoproducts which are toxic.

Response: We have removed the segment “which is less toxic than UV light” from the sentence “DEACM-aa-tRNA was rapidly uncaged by visible light at ~400 nm, which is less toxic than UV light.”

3) Page 7: The Discussion is not very descriptive and does not sufficiently emphasize the novelty of this work.

Response: Per the reviewer’s suggestion, we have rewritten the Discussion to be more descriptive and thorough, and to emphasize the novelty of our findings.

4) Figure 1c: The y-axis is a logarithmic function; this should be shown. The release seems to level off at approximately 30 s (60% conversion). Is the full conversion eventually achieved?

Response: As shown in Figure 2a (left), DEACM-aa-tRNA disappeared after irradiation for more than 40 s, indicating the full conversion of DEACM-aa-tRNA to aa-tRNA. We have added the following sentence in the legend of Figure 1c. “Note that the y-axis is on a logarithmic scale.”

Reviewer #1 (Remarks to the Author)

The revised manuscript by Ohtsuki and colleagues is an important demonstration of photo-triggered control of protein synthesis. The paper is concise, the experiments are well done, the data are rigorously controlled, and the topic is of broad interest. Of note, the authors carefully and directly addressed my previous concerns and the work has been significantly improved. Although I am still not certain of the novelty given the extension of their previous work, the improvements may merit publication. As mentioned in my first review, the idea of spatiotemporally photocontrolling translation has applications for synthetic biology, minimal cells, and optogenetics.

Reviewer #2 (Remarks to the Author)

The authors have adequately addressed my initial concerns and provided convincing experimental data to back up their claims.

Reviewer #1: The revised manuscript by Ohtsuki and colleagues is an important demonstration of photo-triggered control of protein synthesis. The paper is concise, the experiments are well done, the data are rigorously controlled, and the topic is of broad interest. Of note, the authors carefully and directly addressed my previous concerns and the work has been significantly improved. Although I am still not certain of the novelty given the extension of their previous work, the improvements may merit publication. As mentioned in my first review, the idea of spatiotemporally photocontrolling translation has applications for synthetic biology, minimal cells, and optogenetics.

Reviewer #2: The authors have adequately addressed my initial concerns and provided convincing experimental data to back up their claims.

Response: We thank reviewers #1 and #2 for these comments.